# Fluoride Status and Cardiometabolic Health: Findings from a Representative Survey among Children and Adolescents

**DOI:** 10.3390/nu14071459

**Published:** 2022-03-31

**Authors:** Jessica A. Ballantyne, Gemma Coyle, Sneha Sarwar, Tilman Kühn

**Affiliations:** 1Institute for Global Food Security (IGFS), School of Biological Sciences, Queen’s University Belfast, Belfast BT9 5DL, UK; jballantyne01@qub.ac.uk (J.A.B.); gcoyle08@qub.ac.uk (G.C.); 2Institute of Nutrition and Food Sciences, University of Dhaka, Dhaka 1000, Bangladesh; snehasarwar4@gmail.com; 3Heidelberg Institute of Global Health (HIGH), University of Heidelberg, 69120 Heidelberg, Germany

**Keywords:** fluoride, NHANES, blood pressure, HbA1c, lipids, CRP, children, adolescents, C-reactive protein

## Abstract

There is preliminary evidence to suggest a positive association between fluoride exposure and higher blood pressure among children, but population-based biomarker studies are lacking. Thus, data from the 2013/2014 and 2015/2016 cycles of the US-based National Health and Nutrition Examination Survey (NHANES) were analysed to evaluate the association between plasma fluoride concentrations and blood pressure among children and adolescents aged 8 to 19 years. Secondary analyses were carried out on fluoride status in relation to further markers of cardio-metabolic health, i.e., anthropometric indices, biomarkers of lipid and sugar metabolism, and C-reactive protein levels. There was a positive correlation between water fluoride content and plasma fluoride concentrations (Spearman’s r = 0.41, *p* < 0.01). However, multivariable linear regression models did not show significant differences in adjusted mean values of systolic and diastolic blood pressure across increasing quartiles of fluoride concentrations. Further markers of cardio-metabolic health were not associated with fluoride status, with the exception of a weak inverse association between plasma fluoride and HbA1c levels. Higher plasma fluoride may not be a risk factor for increased blood pressure or impaired cardio-metabolic health among children in the USA, a non-fluoride endemic country, with wide-spread water fluoridation.

## 1. Introduction

The controlled fluoridation of drinking water has been implemented in many countries worldwide for the prevention of dental caries [1,2]. While studies show that fluoridation provides benefits regarding oral health, ‘anti-fluoridation’ opinions have been voiced in reference to its putative negative health consequences, such as increased risks for autism, certain cancers, or reduced intelligence [3]. However, a comprehensive review of laboratory-based mechanistic and epidemiological studies suggested that potential health risks related to fluoride exposure may be due to biases of epidemiological studies with low methodological quality [4].

With respect to blood pressure, a recent systematic review of epidemiological studies indicated a positive association between fluoride exposure and elevated blood pressure among individuals from endemic fluorosis areas in Iran, Turkey and China [5], but not in other study populations. Again, the quality of the included study was graded as low because objective biomarkers measurements of fluoride status were lacking and potential confounders were not accounted for in the majority of studies [5]. The only identified study among children was a biomarker study in a well-characterised cross-sectional sample of children aged 10 to 18 years from Mexico City, a non-fluoride endemic area. This study showed that higher plasma fluoride levels were related to higher systolic and diastolic blood pressure levels, and to higher BMI, trunk fat, insulin, and fasting-glucose levels, albeit only among girls [6].

Here, we used data from the 2013/2014 and 2015/2016 cycles of the National Health and Nutrition Examination Survey (NHANES) to evaluate the association between objectively measured fluoride status, i.e., plasma fluoride levels, and systolic and diastolic blood pressure among 3495 children and adolescents aged 8 to 19 years. Considering the above-mentioned finding of a worse pattern of cardio-metabolic parameters among girls from Mexico reported by Liu et al. [6], we further investigated whether higher plasma fluoride was associated with BMI, waist circumference, blood lipids (total, HDL, and non-HDL cholesterol), HbA1c, glucose, insulin, and C-reactive protein levels.

## 2. Materials and Methods

### 2.1. Study Population

The National Health and Nutrition Examination Survey (NHANES) is an ongoing cross-sectional survey carried out in the USA. Since the 1990s, the study provides detailed health status monitoring in large, nationally representative samples of non-institutionalised individuals aged 1 to 80 years, from survey cycles carried out every two years. Assessments include detailed medical interviews, physical examinations, and collections of urine and blood samples for comprehensive biomarker measurements. An overview of the protocols used for different examinations and interview assessments can be found on the NHANES website [7]. All NHANES examinations and the public release of the study’s data were approved by the Research Ethics Review Board (ERB) of the US National Center for Health Statistics (NCHS). Children aged 7–17 years provided informed assent, and parents of these participants provided informed consent. Participants aged 18 years and older provided informed consent.

For the present analyses on fluoride status and parameters of cardio-metabolic health, data from the NHANES cycles 2013–2014 and 2015–2016 were used as plasma fluoride was measured among study participants aged 8 to 19 years in these cycles. Out of the 5451 children and adolescents aged 8 to 19 years, who took part in NHANES between 2013 and 2016, fluoride status was measured among 4470, and three blood pressure measurements were obtained for 3495 of these study participants. Thus, the analytical sample for the present analyses consisted of 3495 individuals. Multiple imputation by Fully Conditional Specification was carried out in case of sporadic missing values for covariates (ratio of family income to poverty: *n* = 59; parental education level: *n* = 116; plasma cotinine: *n* = 146; waist circumference and body mass index: *n* = 25) accounting for the survey design.

The sample sizes were smaller due to missing blood biomarker measurements for analyses on fluoride status and secondary endpoints (blood lipids: *n* = 3443; glucose and insulin: *n* = 2173; HbA1c: *n* = 2212; and C-reactive protein: *n* = 1823).

### 2.2. Fluoride and Blood Pressure Measurements

Details on the methods for fluoride measurements can be found on the NHANES website [8,9]. In short, fluoride levels were quantified in tap water and plasma samples using an ion-specific electrode. The lower limits of detection for fluoride in water and plasma were 0.10 mg/L and 0.25 µmol/L. Inter-assay coefficients of variation based on repeated measurements of fluoride concentrations in low, medium, and high-quality control samples were below 5% across both NHANES cycles in plasma and water. Details on the measurements of anthropometric parameters and further plasma biomarkers (blood lipids, biomarkers of sugar metabolism, and C-reactive protein) can be found online on the NHANES website [7]. The NHANES staff took three consecutive blood pressure measurements, after allowing the participants to rest in a seated position for five minutes. The first measurement was not included in the final calculation of average systolic and diastolic blood pressure as it was taken to allow for any external factors influencing the measurement to be removed. Further details on the equipment and protocols used for the blood pressure measurements are available online [10].

### 2.3. Statistical Methods

Survey-weighted means (standard errors) and survey-weighted frequencies were obtained for continuous and categorical covariates, respectively, across sex-specific quartiles of plasma fluoride levels in order to describe the study population. The age- and sex-adjusted Spearman’s coefficient was calculated to evaluate the correlation between plasma and water fluoride levels. In addition, this correlation was visualised by a bubble plot.

Associations between blood pressure parameters, secondary endpoints (BMI, waist circumference, biomarkers of lipid and sugar metabolism, and C-reactive protein) and sex-specific quartiles of plasma fluoride levels were evaluated by survey-weighted linear regression models, using log2-transformed endpoint parameters as continuous dependent variables. In addition, plasma fluoride levels were modelled on the log2 scale to test for linear trends in associations with blood pressure parameters and secondary endpoints.

Regression models were first adjusted for age and sex (Model 1). Additional multivariable adjustment was carried out for the second multivariable linear regression model for standing height (cm), waist circumference (cm), BMI (kg/m^2^), household reference person’s education level (less than 9th grade, 9–11th grade, high school graduate, some college or AA degree and college graduate or above), cotinine levels (ng/mL) as a biomarker of exposure to second-hand smoke, ethnicity (Mexican American, Other Hispanic, Non-Hispanic White, Non-Hispanic Black, Non-Hispanic Asian, and Other Race including multi-racial), ratio of family income to poverty, and fasting duration (minutes). Confounders were selected by literature search.

As the above-mentioned study among Mexican children had shown sex differences in associations between fluoride levels and blood pressure parameters as well as anthropometric indices and blood lipids [6], we carried out regression analyses stratified by sex in addition to our main analyses. Associations were considered as statistically significant at two-sided *p*-values < 0.05. Statistical analyses were carried out using SPSS 26.0.0.0 (IBM, Armonk, NY, USA).

## 3. Results

### 3.1. Characteristics of the Study Population

Characteristics of the study population are shown in Table 1. Mean ages were 14.0 ± 0.1, 13.4 ± 0.1, 13.4 ± 0.1 and 13.8 ± 0.2 years across quartiles of plasma fluoride. The proportion of Non-Hispanic White participants was slightly greater in the highest fluoride quartile (60.2% vs. 50.1% in the lowest quartile), whereas the proportion of Mexican Americans showed an opposite trend (12.8% vs. 18.4%). The percentage of household reference persons with higher education level (28.6% with at least a college degree) was higher in fluoride quartile 4 compared to quartile 1 (22.7%). 

While most other characteristics showed similar proportions and mean values across quartiles of plasma fluoride, there were tendencies for higher triglyceride (Q4: 102.7 ± 4.3 vs. Q1: 94.2 ± 3.1 mg/dL), water fluoride (Q4: 0.67 ± 0.04 vs. Q1: 0.30 ± 0.03 mg/L) and cotinine levels (Q4: 12.7 ± 2.5 vs. Q1: 4.5 ± 1.0 ng/L) with higher fluoride levels. Fasting duration was slightly shorter in the highest compared to the lowest quartile (Q4: 362.1 ± 13.8 min vs. Q1: 399.2 ± 13.9). There was a statistically significant positive correlation between water fluoride content and plasma fluoride concentrations (Spearman’s r = 0.41, *p* < 0.01, also visualised in Figure A1).

### 3.2. Associations between Fluoride Status and Cardio-Metabolic Biomarkers

Adjusted mean values of blood pressure across quartiles of plasma fluoride are shown in Table 2. There were no statistically significant associations between fluoride levels and systolic or diastolic blood pressure, neither in the age- and sex-adjusted regression model nor in the multivariable-adjusted one.

Results from regression analyses on fluoride status and further cardiometabolic endpoint parameters are shown in Table A1. With regard to BMI and waist circumference, age- and sex-adjusted regression analyses did not show significant associations. In the multivariable-adjusted model, there were non-significant trends for positive associations with both parameters, with *p*-values for linear trend of 0.06 and 0.07, respectively. The only parameter for which a significant linear association with fluoride was observed in both regression models was HbA1c, with marginally lower values in the highest quartile (5.2% vs. 5.3% in the lowest quartile). Results from ancillary analyses stratified by sex are presented in Table A2. The above-mentioned inverse association between fluoride and HbA1c levels was statistically significant among female participants (p linear trend = 0.007) but not among male participants (p linear trend = 0.11). Moreover, there was a borderline significant inverse association between fluoride and HDL–cholesterol levels among female study participants (p linear trend = 0.04), whereas HDL was not associated with fluoride among male participants.

## 4. Discussion

In the present study, fluoride exposure objectively measured by plasma fluoride levels was not associated with systolic and diastolic blood pressure in a large study among 8- to 19-year-old children and adolescents from the USA. A statistically significant inverse association was observed between plasma fluoride levels and HbA1c values, although this association was weak in magnitude. Plasma fluoride was not associated with other biomarkers of cardio-metabolic state.

One motivation for the present analyses was a recent study by Liu et al., in which higher plasma fluoride was significantly associated with blood pressure among girls from Mexico City [6]. The reason for the discrepancy in the results by Liu et al. and those from the present study remain unclear, as average plasma fluoride concentrations in the study from Mexico were at 0.21 µmol/L, i.e., lower than in the present one (0.40 µmol/L). This difference in plasma fluoride is consistent with the fact that community water fluoridation is not common in Mexico, unlike in the USA, where around 73% of the population consume fluoride-enriched water [6,11]. Additionally, it should be noted that the significant associations in the study by Liu et al. were only evident among girls but not among boys [6], while this analysis showed no indication for a sex difference. Thus, further studies are needed to provide more clarity on potential associations between fluoride status and blood pressure among children and adolescents. Among adults, fluoride exposure has been associated with increased blood pressure in fluoride endemic areas [5,12], while biomarker studies from other areas are lacking and are needed before conclusions on a potential blood pressure increasing effect of moderate fluoride exposure can be drawn.

Unlike Liu et al., who reported higher glucose and insulin levels among girls with higher plasma fluoride levels [6], we did not observe significant associations between fluoride status and these biomarkers. By contrast, there was a moderate inverse association between fluoride and HbA1c among female participants of the present study. We further found an inverse association between plasma fluoride and HDL-cholesterol, while no such association was detected by Liu et al. [6]. The present study and the study by Liu et al. were consistent in that they showed borderline significant tendencies for positive associations between higher fluoride levels, BMI, and waist circumference, although it should be noted that both studies were cross-sectional. Therefore, reverse causality, i.e., higher fluoride intake via food and water among obese children, cannot be ruled out, and prospective studies are needed to investigate a potential obesogenic effect of fluoride exposure.

While this study indicates that plasma fluoride levels are not related to increased cardio-metabolic risks among children and adolescents, we acknowledge that analyses from the same NHANES dataset suggest associations between fluoride and increased levels of biomarkers of kidney and liver function [13] and sex steroid hormones [14]. Moreover, higher plasma fluoride levels were associated with sleep disturbances in NHANES [15], which is in line with findings from a recent cross-sectional study among Canadian adults [16]. Again, prospective studies are needed to better assess temporal relationships between fluoride exposure and these health outcomes, while the mentioned studies do underline the necessity for an integrated risk-benefit assessment of fluoride exposure taking into account a broad range of health indicators. Interestingly, we found a tendency for higher cotinine levels among study participants with higher plasma fluoride, which is in line with previous studies that suggest that tobacco exposure may increase fluoride levels [17,18]. Thus, accounting for smoking as a potential confounder seems important for future analyses of potential adverse health effects of fluoride.

The following methodological strengths and limitations need to be considered when interpreting the present findings. NHANES is representative for the US general population, and importantly, objective biomarkers of fluoride exposure were measured. The sample size facilitated well-powered statistical analyses, and data on important potential confounders such as smoking exposure were available. However, the cross-sectional design of the study needs to be acknowledged. A potential further limitation of this research project is the focus on two NHANES cycles (2013/2014 and 2015/2016), while biomarker studies from other areas of the world are lacking and needed. Finally, the statistically significant associations between fluoride and HbA1c and HDL should be interpreted with caution, given the moderate differences in mean biomarker concentrations across quartiles of fluoride levels and the high number of statistical tests. 

## 5. Conclusions

The present study does not point to fluoride-related cardio-metabolic health risks among children and adolescents in non-fluoride endemic regions where moderate community water fluoridation is common. Nevertheless, in view of cross-sectional studies that suggest that a higher fluoride exposure may be related sleep problems and increased levels of biomarkers of kidney and liver function as well steroid hormone metabolism, further and particularly prospective biomarker studies are needed on fluoride in relation to a broader range of health outcomes, not only among children and adolescents but also among adults.

## Figures and Tables

**Table 1 nutrients-14-01459-t001:** Characteristics of the study population across quartiles of plasma fluoride (*n* = 3495) *.

	Plasma Fluoride (Range in µmol/L)			
	Quartile 1	Quartile 2	Quartile 3	Quartile 4
	(0.18–0.26)	(0.24–0.34)	(0.33–0.47)	(0.45–16.67)
Age (years)	14.0 ± 0.1	13.4 ± 0.1	13.4 ± 0.1	13.8 ± 0.2
Female, *n* (%)	435 (49.4)	429 (52.2)	463 (54.0)	435 (50.5)
Male, *n* (%)	468 (50.6)	444 (47.8)	411 (46.0)	410 (49.5)
Education Level, *n* (%) **				
Less than 9th Grade	138 (12.6)	105 (8.1)	90 (7.5)	68 (5.1)
9–11th Grade	146 (13.8)	124 (11.1)	121 (11.0)	125 (12.6)
High School Graduate	182 (19.1)	178 (20.0)	217 (22.6)	201 (21.0)
Some College or AA Degree	260 (31.9)	296 (37.2)	247 (30.7)	275 (32.8)
College Graduate or above	177 (22.7)	170 (23.6)	199 (28.2)	176 (28.6)
Ethnicity, *n* (%)				
Mexican American	252 (18.4)	224 (17.8)	189 (14.6)	163 (12.8)
Other Hispanic	119 (9.5)	115 (9.7)	109 (8.0)	66 (5.0)
Non-Hispanic White	212 (50.1)	206 (49.0)	248 (55.6)	279 (60.2)
Non-Hispanic Black	165 (11.4)	192 (13.3)	217 (13.4)	213 (13.0)
Non-Hispanic Asian	95 (5.1)	87 (5.4)	61 (3.4)	73 (3.9)
Other	60 (5.5)	49 (4.9)	50 (5.1)	51 (5.1)
Ratio of family income to poverty	2.33 ± 0.11	2.48 ± 0.13	2.50 ± 0.15	2.45 ± 0.13
Body Mass Index (kg/m^2^)	23.0 ± 0.3	22.4 ± 0.2	22.9 ± 0.3	22.9 ± 0.4
Standing Height (cm)	160.8 ± 0.6	157.4 ± 0.8	156.5 ± 0.6	157.6 ± 0.8
Fasting duration (minutes)	399.2 ± 13.9	416.0 ± 19.4	386.4 ± 13.1	362.1 ± 13.8
Fluoride, Water (mg/L)	0.30 ± 0.03	0.46 ± 0.04	0.55 ± 0.03	0.67 ± 0.04
Fluoride, Plasma (µmol/L)	0.19 ± 0.001	0.29 ± 0.001	0.39 ± 0.001	0.70 ± 0.02
Cotinine, serum (ng/mL)	4.5 ± 1.0	3.8 ± 0.8	8.8 ± 2.6	12.7 ± 2.5
Systolic Blood Pressure (mm/Hg)	107.4 ± 0.5	106.5 ± 0.5	106.6 ± 0.5	106.5 ± 0.6
Diastolic Blood Pressure (mm/Hg)	58.4 ± 0.7	57.2 ± 0.5	57.1 ± 0.7	57.6 ± 0.9
Glucose (mg/dL)	89.5 ± 1.3	89.2 ± 0.7	90.1 ± 0.9	89.5 ± 0.6
HbA1c (%)	5.3 ± 0.19	5.2 ± 0.21	5.2 ± 0.02	5.2 ± 0.02
Insulin (uU/mL)	14.3 ± 1.3	12.9 ± 1.0	14.1 ± 1.0	13.5 ± 0.8
Triglycerides (mg/dL)	94.2 ± 3.1	95.9 ± 3.6	105.6 ± 5.5	102.7 ± 4.3
HDL-Cholesterol (mg/dL)	53.1 ± 0.9	54.5 ± 0.6	52.9 ± 0.6	52.1 ± 0.7
Total Cholesterol (mg/dL)	155.1 ± 1.4	156.8 ± 1.0	156.2 ± 1.5	154.5 ± 1.1
Non-HDL Cholesterol (mg/dL)	102.0 ± 1.5	102.4 ± 1.0	103.3 ± 1.4	102.3 ± 1.3
C-Reactive Protein (mg/L)	1.63 ± 0.23	1.55 ± 0.18	1.58 ± 0.14	1.55 ± 0.27

Frequencies for categorical variables and mean values ± standard errors (SE) for continuous variables (survey-weighted). * Sample sizes smaller for the following biomarkers: blood lipids: *n* = 3443; glucose and insulin: *n* = 2173; HbA1c: *n* = 2212; and C-reactive protein: *n* = 1823. ** of household reference person.

**Table 2 nutrients-14-01459-t002:** Adjusted mean levels of blood pressure and metabolic biomarkers across sex-specific quartiles of plasma fluoride (*n* = 3495).

	Quartile 1	Quartile 2	Quartile 3	Quartile 4	*p* _ *trend* _
Regression Model 1					
Systolic Blood Pressure (mm/Hg)	106.6 ± 0.4	106.3 ± 0.5	106.3 ± 0.4	105.9 ± 0.6	0.43
Diastolic Blood Pressure (mm/Hg)	57.6 ± 0.6	57.0 ± 0.5	57.1 ± 0.6	57.0 ± 0.9	0.55
Regression Model 2					
Systolic Blood Pressure (mm/Hg)	106.8 ± 0.4	106.6 ± 0.4	106.4 ± 0.4	106.1 ± 0.6	0.41
Diastolic Blood Pressure (mm/Hg)	57.1 ± 0.7	56.8 ± 0.5	56.8 ± 0.65	56.7 ± 0.9	0.60

Mean values and standard errors obtained from survey-weighted multivariable linear regression models by the least squares means method; Model 1 adjusted for age and sex; Model 2 further adjusted for serum cotinine levels, BMI, waist circumference, height, education level of the household’s reference person, race/ethnicity, poverty-to-income ratio, and fasting duration; *p* values for linear trend from linear regression models with blood pressure parameters on the log2 scale as the dependent variable and plasma fluoride concentrations on the log2 scale as the independent variable.

## Data Availability

Data of the National Health and Nutrition Examination Survey including those used for the present study can be downloaded on the homepage of the Centers of Disease Control and Prevention (https://wwwn.cdc.gov/nchs/nhanes/Default.aspx, accessed on 1 December 2020).

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
