# Peer review of "Fluoride Status and Cardiometabolic Health: Findings from a Representative Survey among Children and Adolescents"

_nutrients, 2022, doi:10.3390/nu14071459_

Round 1

Reviewer 1 Report

This study has some interesting data but the authors failed to present their study in a fascinating way. Most of the results are presented in table format. I strongly suggest authors to present their results in a different way. Also, re-write the results section because it is hard to follow.

The conclusion of study needs to be highlighted. 

Reviewer 2 Report

The reviewed manuscript entitled „Fluoride Status and Cardiometabolic Health: Findings from a Representative Survey among Children and Adolescents” written by Jessica A. Ballantyne et al. presents an interesting attempt to explore relationships between plasma fluoride levels and blood pressure in individuals aged 8 to 19 years. Associations with cardio-metabolic biomarkers were also examined. This paper makes a significant contribution to the discussion of the impact of water fluoridation on health. The reviewed manuscript is scientifically sound, well organized, and English language is correct. The manuscript could also be improved on some points, and my suggestions for improvements are addressed in the comments below.

General concept comments:

  1. I have one comment on the general concept of the analysis performed for this study. The authors performed multiple regression models, adjusted by sex, age (model 1) and other characteristics (model 2). It could be very interesting to perform additional multivariate regression modelling using all analyzed variables to select the best model (ex. using stepwise regression) explaining plasma fluoride levels. The variables retained in the best model could give more insight into the examined relationships and make the results more complete.

Specific comments:

  1. The text includes the expressions “HbA1” and “HbA1c”. The difference between them should be provided; however, it seems that both of these expressions mean glycated hemoglobin; therefore, they should be unified by changing “HbA1” to “HbA1c”.
  2. In lines 79-82, the information about performed imputation was provided. Please clarify what method was used for the imputation.
  3. A moderate, statistically significant correlation was revealed between the water fluoride content and plasma fluoride concentrations (lines 137-139). It could be beneficial to graphically show this relationship by providing a scatterplot for these two related variables together with a trend line (ex. simple linear regression line). Such a plot could be placed in the Appendix section.
  4. In line 152, it seems that instead “are presented in Appendix A, Tables A1” should be “are presented in Table A1”.
  5. Paragraph 3.1 contains an interpretation of the most interesting results presented in Table 1. Despite the variables included in this paragraph, there are two more variables with quite different values between the analyzed quartiles: education level less than 9th grade and proportion of Mexican American. These characteristics seem to occur more frequently in quartiles with lower plasma fluoride concentrations and could be especially interesting in relation to the background of the Liu et al. article. Interpretation of these results could be added to 3.1 paragraph. To better understand the differences in the analyzed characteristics between the quartiles, a new column with p values resulting from appropriate statistical testing of these differences could be added to Table 1.
  6. Table A1 needs to be improved. There are really two Tables A1, first of them is lacking of footnote, the second is lacking of data. Please correct this table.

I believe that my suggestions will be helpful to the authors in increasing the quality of the reviewed paper.
